# Injectable Hyaluronan-Based Thermoresponsive Hydrogels for Dermatological Applications

**DOI:** 10.3390/pharmaceutics15061708

**Published:** 2023-06-11

**Authors:** Si Gou, Alexandre Porcello, Eric Allémann, Denis Salomon, Patrick Micheels, Olivier Jordan, Yogeshvar N. Kalia

**Affiliations:** 1School of Pharmaceutical Sciences, University of Geneva, 1211 Geneva, Switzerland; si.gou@unige.ch; 2Institute of Pharmaceutical Sciences of Western Switzerland, University of Geneva, 1211 Geneva, Switzerland; 3KYLYS Sàrl, 34, Route de la Galaise, c/o FONGIT, Plan-les-Ouates, 1228 Geneva, Switzerland; alexandre.porcello@kylys.com (A.P.); eric.allemann@kylys.com (E.A.); olivier.jordan@kylys.com (O.J.); 4Clinique Internationale de Dermatologie Genève SA, 1201 Geneva, Switzerland; denis.salomon@cidge.ch; 5Private Practice, 8, Chemin de la Fontaine, Chêne-Bougeries, 1224 Geneva, Switzerland; patrickscab@bluewin.ch

**Keywords:** dermal fillers, hyaluronic acid, thermoresponsive, poly(N-isopropylacrylamide), ex vivo skin model

## Abstract

Most marketed HA-based dermal fillers use chemical cross-linking to improve mechanical properties and extend their lifetime in vivo; however, stiffer products with higher elasticity require an increased extrusion force for injection in clinical practice. To balance longevity and injectability, we propose a thermosensitive dermal filler, injectable as a low viscosity fluid that undergoes gelation in situ upon injection. To this end, HA was conjugated via a linker to poly(N-isopropylacrylamide) (pNIPAM), a thermosensitive polymer using “green chemistry”, with water as the solvent. HA-L-pNIPAM hydrogels showed a comparatively low viscosity (G′ was 105.1 and 233 for Candidate1 and Belotero Volume^®^, respectively) at room temperature and spontaneously formed a stiffer gel with submicron structure at body temperature. Hydrogel formulations exhibited superior resistance against enzymatic and oxidative degradation and could be administered using a comparatively lower injection force (49 N and >100 N for Candidate 1 and Belotero Volume^®^, respectively) with a 32G needle. Formulations were biocompatible (viability of L929 mouse fibroblasts was >100% and ~85% for HA-L-pNIPAM hydrogel aqueous extract and their degradation product, respectively), and offered an extended residence time (up to 72 h) at the injection site. This property could potentially be exploited to develop sustained release drug delivery systems for the management of dermatologic and systemic disorders.

## 1. Introduction

Injectable biomaterials have become a hot topic as they have been used to address problems associated with aging skin, e.g., decreased skin elasticity, facial wrinkles, and collagen degradation [1]. Dermal fillers can achieve rapid facial rejuvenation through simple and short procedures and have been widely investigated [2,3]. Among the different materials, hyaluronic acid (HA)-based fillers account for the largest number of products with more than 2.6 million minimally invasive procedures conducted in the US alone in 2020 [4,5]. This can be attributed to the non-toxicity of HA, its biocompatibility and easy reversibility [6,7].

Natural HA is composed of repeating alternating units of D-glucuronic acid and N-acetyl-D-glucosamine, all connected by “β-linkages, GlcA β (1→3) GlcNAc β (1→4) [8]. It is synthesized as a compound of high molecular weight in both the epidermis and dermis but rapidly undergoes degradation induced by oxidative stress and hyaluronidases. The half-life (t_1/2_) of unmodified HA in the skin is about 12 h [9,10,11,12,13,14]. The fragmented HA of lower molecular size is a potent inducer of inflammation and angiogenesis, which can elicit adverse effects such as erythema, slight oedema, hematoma, itching, and pain [15,16,17,18,19,20]. Therefore, HA used in dermal fillers is routinely cross-linked to improve mechanical properties and in vivo residence time [21,22,23]. It was reported that the lifetime of the filler material is dependent on the type and density of the cross-linking agents [24]: 1,4-butanediol diglycidyl ether (BDDE) is the most widely used crosslinker in commercial HA dermal fillers, and a high cross-linking degree generally results in a stiffer product with higher elastic modulus (G′) that potentially extends the residence time in vivo.

However, the high cross-linking degree was associated with hypersensitivity reactions since the unreacted cross-linking agents or their by-products could be toxic [25,26,27], and the increased number of modifications would possibly lead to decreased biocompatibility of HA-based materials as the reaction conditions (e.g., heat, alkaline conditions) are prone to degrade HA gels and release small HA fragments with potential safety issues [28,29,30,31]. In addition, the increased extrusion force required for intradermal injection can be an issue in clinical practice. It is crucial yet complicated to develop a biocompatible product with an optimal balance between biocompatibility, prolonged residence time and injectability. To date, issues reported with HA-based dermal filler products on the market include difficulties in precise injection, unsatisfying/suboptimal volumizing effect and the need for repeated injections [32].

Recently, a novel technology that involved a specific conjugation of the thermoresponsive polymer, poly(N-Isopropylacrylamide) (pNIPAM), to the linear HA backbone via a cyclooctyne linker (L) was reported [33,34]. Due to the desolvatation of the hydrophobic pNIPAM moieties above a defined lower critical solution temperature (LCST) and to the presence of the linker, the synthetic HA-L-pNIPAM copolymer spontaneously forms submicron spherical particles above the LCST [33,34,35,36]. These domains act as inter-chain crosslinkers, resulting in a sol–gel copolymer transition driven only by physical interactions, avoiding the use of chemical cross-linking agents. Preliminary research has shown the formation of microgel structures upon subcutaneous injection in mice that precisely confine the HA-L-pNIPAM hydrogel at the injection site, offering an extended residence time with a lower risk of migration [37].

In this study, the in situ physical crosslinking technology was employed to design products for a durable and precisely controlled skin correction. The specific aims were (i) to develop a new family of HA-L-pNIPAM copolymers with optimized physical characteristics as implantable biomaterials for dermatological application in terms of rheological properties, cohesivity, injectability, and in vitro resistance to oxidative stress and hyaluronidase-mediated degradation, (ii) to evaluate cytocompatibility of the candidate formulations using the L929 cell line in vitro, and (iii) to investigate their subcutaneous distribution pattern as a function of time after implantation in an ex vivo porcine skin model.

## 2. Materials and Methods

### 2.1. Materials

Laboratory-grade hyaluronic acid (HA) sodium salt (1500–1750 kDa) was purchased from Contipro a.s. (Dolní Dobrouč, Czech Republic). Sulfo-Dibenzocyclooctyne-PEG_4_-amine (Sulfo DBCO-PEG4-NH2) was bought from Click Chemistry Tools (Scottsdale, AZ). N-(3-dimethylaminopropyl)-N’-ethylcarbodiimide (EDC), N-hydroxysuccinimide (NHS), azide-terminated poly(N-isopropylacrylamide) (pNIPAM-N_3_; 15 kDa), hyaluronidase from bovine testes (Type VI-S), hydrogen peroxide (30% *w*/*w*), toluidine blue, Dulbecco’s Modified Eagle’s Medium—high glucose (DMEM), embryoMax L-glutamine solution (100X), foetal bovine serum (FBS), and the antibiotic–antimycotic solution (100X) were purchased from Sigma-Aldrich (St Louis, MO, USA). Dialysis membranes (Biotech CE Dialysis Tubing 300 kDa) were acquired from Repligen (Waltham, MA, USA). The Alcian blue PAS stain kit was purchased from Abcam (Cambridge, UK). Belotero Balance^®^ and Belotero Volume^®^ were purchased from Merz Pharma (Geneva, Switzerland) and used as reference products.

### 2.2. Synthesis of HA-L-pNIPAM Copolymers

HA-L-pNIPAM copolymers (HA-L-pNIPAM_0.10_, HA-L-pNIPAM_0.25_, HA-L-pNIPAM_0.50_) were synthesized using a slightly modified procedure based on a previous publication [33,34]. The scheme for the green chemistry synthetic route of the HA-L-pNIPAM copolymers is presented in Appendix A. Briefly, HA (1500–1750 kDa) was solubilized in distilled water at 0.2% (*w/v*) under magnetic stirring. After dissolution of HA, EDC (5 eq. COO^−^) and NHS (5 eq. COO^−^) were added at intervals of 15 min. The pH was adjusted to 5.5 using a 0.1 M NaOH and a 0.1 M HCl and monitored using a Metrohm pH gel electrode (Herisau, Switzerland). Then, Sulfo-DBCO-PEG4-amine, previously dissolved in distilled water, was added (0.1 eq. COO^−^ for HA-L-pNIPAM_0.10_, 0.25 eq. COO^−^ for HA-L-pNIPAM_0.25_, and 0.5 eq. COO^−^ for HA-L-pNIPAM_0.50_) and was stirred overnight (12 h) at room temperature for amidation. The intermediate product (HA Sulfo DBCO-PEG4) was dialyzed thrice against a 5% (*w/v*) NaCl solution (MWCO: 300,000, 3 h, at ambient temperature) and then three times against distilled water before being transferred to a round bottom flask. Then, the DBCO group of the linker reacted via copper-free azide-DBCO click chemistry with pNIPAM-N_3_ (15 kDa, 1 eq. DBCO). The pH was adjusted to 7 using a 0.1 M NaOH and a 0.1 M HCl. The reaction was allowed to proceed for 12 h under stirring at room temperature. The final product was dialyzed thrice against a 5% (*w/v*) NaCl (MWCO: 300,000, 3 h, ambient temperature) and three times against distilled water before being frozen at −80 °C, lyophilized (Freeze Dryer Alpha 1–4 LD plus, Christ, Osterode am Harz, Germany; 48 h, 1.5·10^−1^ mbar, −80 °C) and stored at 4 °C.

The intermediate product (5 mg) and the final product (10 mg) were digested with hyaluronidase (300 IU/mL) using D_2_O as a solvent; then, their chemical structures and degrees of substitution (DS) were determined from the ^1^H NMR spectra acquired on a Bruker Avance Neo 600 MHz NMR spectrometer at ambient temperature. The DS was calculated using the integration ratio of methyl protons of HA (δ 2.00 ppm) with the aromatic protons of the DBCO group (7.67 ppm) for the intermediate product (DS_1_) and the integration of the methyl protons of pNIPAM (δ 1.13) with the aromatic protons of the DBCO group (7.67 ppm) for the final product (DS_2_).

### 2.3. Physical Characterization of HA-L-pNIPAM Hydrogels

#### 2.3.1. Rheological Properties

Rheological behaviors were determined on a HAAKE Mars Rheometer™ (Thermo Scientific, Waltham, MA, USA) equipped with a Peltier cone-plate C35 2°/Ti rotor. Measurements were performed on 420 µL samples with a sample hood to reduce evaporation. First, the synthetic HA-L-pNIPAM copolymers were dissolved in PBS under agitation overnight; then, centrifugation was performed at 10,000 rpm for 20 min at 4 °C. The storage modulus (G′) and loss modulus (G″) of the obtained formulations (3%, *w*/*v*) were assessed as a function of temperature using a ramp from 22 °C to 37 °C with a heating rate of 0.04 °C/s and a constant oscillatory frequency of 0.7 Hz, simulating the forces to which a filler is exposed in vivo from gravity and muscular movements [38].

Then, three HA-L-pNIPAM hydrogel candidates were selected for further physical and biological characterization. Briefly, Candidate 1 (2%, *w*/*v*) and Candidate 2 (2%, *w*/*v*) were formulated by directly dissolving the synthetic HA-L-pNIPAM_0.10_ and HA-L-pNIPAM_0.25_ in PBS, respectively, whereas Candidate 3 (2%, *w*/*v*) was prepared as a combination of HA-L-PNIPAM_0.50_ and non-derivatized linear HA (1500–1750 kDa) with a ratio of 3:1 (*w*/*w*). The G′ and G″ values of the three HA-L-pNIPAM hydrogel candidates and the two reference products (Belotero Balance^®^ and Belotero Volume^®^) were also determined at 22 °C and 37 °C with a constant oscillatory frequency of 0.7 Hz. Shear stress was set to 1.0 N/m^2^ in all experiments to remain in the linear viscoelastic region (LVE).

#### 2.3.2. Injectability

The injection force profile of the three HA-L-pNIPAM hydrogel candidates and the two reference products (Belotero Balance^®^ and Belotero Volume^®^) were determined using a Texture Analyzer TA.XT. Plus (Stable Microsystems Ltd., Surrey, UK). A total of 300 µL of each sample was placed in a 1 mL syringe (Schott TOPPAC^®^, 5 mm internal diameter, with Luer-Lok™, Schott, Mainz, Germany) and extruded through needles of 30G, 32G, and 34G (13 mm, Needle Concept, Biarritz, France) at a speed of 2 mm·s^−1^ at 22 °C, respectively. The maximum force (N) allowed for extruding the samples was set to 100 N.

#### 2.3.3. Accelerated Degradation Assays

The evolutive rheological behaviors of the HA-L-pNIPAM candidate formulations and the commercial reference products under oxidative stress and enzymatic degradation were determined on a HAAKE Mars Rheometer™ equipped with a Peltier cone-plate C35 2°/Ti rotor. To generate oxidative stress, 100 µL of hydrogen peroxide 30% (*w/w*) was added to 400 µL of each sample. The elastic modulus (G′), viscous modulus (G″) and tangent delta (δ) values were measured as a function of time during 12 min at a constant oscillatory frequency of 0.7 Hz. The delay between the addition of H_2_O_2_ and the first measurement was two minutes. For accelerated enzymatic degradation, 100 µL of hyaluronidase (100 U/mL) was added to 400 µL of each sample. Enzyme-mediated degradation was carried out using the same method as describe above. As a control condition, instead of H_2_O_2_ or hyaluronidase, 100 µL of the PBS buffer was added to 400 µL of the sample and assessed at the beginning of the experiments. A sample hood was used during all the measurements to minimize evaporation. Shear stress was set to 1.0 N/m^2^ in all experiments so as to ensure that the measurements were performed in the linear viscoelastic region (LVE).

### 2.4. Biological Evaluations of HA-L-pNIPAM Hydrogels

#### 2.4.1. In Vitro Biocompatibility Evaluation Using a L929 Cell Line

**Sample preparation.** HA-L-pNIPAM candidate formulations (aqueous extracts) and their enzymatic degradation products were tested for cytocompatibility. The extraction dilution method was chosen for the preparation of nondegraded polymer samples as described by the norm ISO 10993-5 regulation. Briefly, the extraction procedure was carried out in the complete medium containing MEM, 10% FBS, a 1% antibiotic–antimycotic solution. A total of 0.5 mL of each formulation was dissolved in a 5 mL medium under continuous agitation at 37 °C for 24 h to obtain an aqueous extract. Degradation products were prepared by incubating 1 mL of each candidate formulation with 100 µL hyaluronidase (100 IU/mL, type I–S, Sigma) at 37 °C for 4 h, followed by heat inactivation. Raw HA (1500–1750 kDa, 2%, *w/v*) was also tested as a non-derivatized control.

**Cytotoxicity test**. L929 cell line (murine fibroblast) was purchased from Sigma-Aldrich (St Louis, MO, USA) and grown in culture flasks containing minimum essential medium (MEM), supplemented with a 10% foetal bovine serum (FBS), a 1% antibiotic–antimycotic solution at 37 °C in a humidified 5% CO_2_ atmosphere and monitored daily using an inverted microscope. Subcultures were performed twice a week when a confluence of 80% was observed. For cytotoxicity tests, cells were seeded into 96-well culture plate (flat bottom, Costar, Corning, Inc.) at a density of 1 × 10^5^ cells per well. After 24 h, the culture media were replaced with a 100 µL sample; a negative control (only MEM with 10% FBS) and a positive control (0.1% SDS solution) were included to validate the viability protocols. The assay was carried out in triplicate. All plates were incubated at 37 °C in a humidified 5% CO_2_ atmosphere, and a WST-1 assay (Roche Applied Science) was performed after 24 h of incubation to quantitatively determine cell viability as previously described [35].

#### 2.4.2. Evaluation of Tissue Integration in Ex Vivo Porcine Skin Model

**Ex vivo porcine skin model.** Porcine ears were obtained from a local slaughterhouse (Loëx, Switzerland) and used within approximately 4 h of sacrifice. The ears were thoroughly cleaned first with running water, then with soap to eliminate microorganisms present on the surface. The skin surface was then shaved and disinfected three times with an aseptic solution (3 × 90 s). The front root part of the ear, which connects to the head and contains abundant fat tissue, was first cut out using a scalpel; then, skin biopsies were performed using a 16 mm diameter punch (Berg & Schmid HK 500; Urdorf, Switzerland) and anchored in a Millicell^®^ cell culture insert (*d* = 12 mm; Merck KGaA, Darmstadt, Germany) with the epidermis facing up. The inserts were positioned in a 12-well plate filled with 3 mL of culture media in each well. The details of culture medium are listed in Appendix A. All manipulations were performed in a 30 cm radius of a flame, with the operator wearing a face mask, gloves and over-blouse to limit airborne contamination. The tools were disinfected with an alcohol solution and flame. The viability of the ex vivo model was previously reported as being up to 72 h by histological observation (H&E staining). In this study, skin biopsies without injection (serving as the blank control group) were harvested at T_0_ and T_72_ and subjected to H&E staining for the evaluation of structural integrity.

**Subcutaneous injection.** The culture plate was incubated at 37 °C and a 5% CO_2_ for 2 h allowing the skin tissue to reach an equilibrium state. The slightly oversized biopsies (*d* = 16 mm) were immobilized in the insert (*d* = 12 mm), which facilitated the subcutaneous injection. A total of 50 µL of each candidate formulation and reference product (raw linear HA and commercial reference) was injected ex vivo through the skin into the subcutaneous fat compartment using a BD Luer-Lok™ 1 mL syringe with a 30G needle. The penetration depth was controlled to be approx. 4 mm and injections were performed by the same researcher under the same aseptic conditions described above; an infrared lamp was used to maintain the ambient temperature at ~40 °C. The skin biopsies were incubated at 37 °C and a 5% CO_2_ and the culture media were changed every 24 h.

**Histology analysis.** Skin biopsies that received candidate formulations were collected at pre-determined time points T_0_ (immediately after injection) and T_72_, while those implanted with raw linear HA and commercial products were harvested at T_0_ and served as control for the validation of histological staining (H&E and AB/PAS). The protocols are presented in Appendix A. Each sample was embedded in OCT and snap-frozen in isopentane chilled with dry ice (−72 °C). In the frozen state, a series of cross-sections with a thickness of 20 µm were obtained using a cryotome (Thermo Scientific CryoStarTM NX70; Reinach, Switzerland). Lamellae containing the implanted materials under visual observation were then placed on Superfrost^®^ positively charged glass slides. The blade and sample temperatures were −35 °C and −25 °C, respectively. The slides were stained following the protocol and were mounted using the Eukitt^®^ mounting medium with a cover slide on the top. Observation was performed in the Bioimaging Core Facility (Faculty of Medicine, University of Geneva) with the Axioscan Z1 brightfield microscope in automatic scan mode.

## 3. Results and Discussion

### 3.1. HA-L-pNIPAM Copolymer Synthesis

The HA-L-pNIPAM copolymers were synthesized via green chemistry synthetic routes given the aqueous solubility of the sulfonated DBCO-PEG_4_-NH_2_ linker [33,34]. The final yield was ~70–80%. The structural characteristics of the HA-L-pNIPAM copolymers were investigated by ^1^H NMR spectroscopy; the spectra are presented in Appendix A. The aromatic protons of DBCO enabled the calculation of the DS_1_ of the amidation from the NMR spectra, with mean values of 1.1, 2.8 and 6.6% for HA-L-pNIPAM_0.10_, HA-L-pNIPAM_0.25_, and HA-L-pNIPAM_0.50_, respectively. These results agree with those of previous reports indicating that amidation of HA with EDC/NHS in water led to a relatively low DS [9,39], and that the DS_1_ can be modulated by adjusting the amount of linker. The DS_2_ for the azide-DBCO grafting with the CH_3_ groups of pNIPAM units was ~90% for the three HA-L-pNIPAM copolymers; the high substitution degree benefited from the hydrophilicity and flexibility of the PEG spacer present in Sulfo DBCO-PEG_4_-NH_2_. The content of each component in the final products calculated from the DS is presented in Appendix A.

### 3.2. Physicochemical Characterization of HA-L-pNIPAM Hydrogels

#### 3.2.1. Rheological Properties

Specific mechanical properties (e.g., stiffness, high G′ values > 50 Pa) are required for injectable volumizing dermal filling materials. Rheological properties of the synthetic HA-L-pNIPAM copolymers were experimentally characterized by dynamic viscoelasticity tests. HA-L-pNIPAM_0.10_, HA-L-pNIPAM_0.25_ and HA-L-pNIPAM_0.50_ were simply dissolved in PBS by agitating overnight, and homogeneous formulations (3%, *w/v*) were obtained after centrifugation. The temperature dependence of their storage modulus (G′) and the loss modulus (G″) are presented in Figure 1.

The three tested HA-L-pNIPAM formulations were all shown to undergo a sharp increase in storage modulus (G′) values and a decrease in loss modulus (G″) values starting from ~27 °C. This shift occurred within a limited range of less than 2 °C, regardless of the DS. The G″ value reached a plateau at ~36 °C, while the G′ value increased continuously at higher temperatures. During all the analyses, the G′ value remained superior to the G″ value (tan δ < 1), suggesting that all the tested formulations presented a gel state in the temperature range of the study. The results confirmed the thermosensitive behavior of the tested formulations, and the observed thickening effect (viscosity increase) supports the hypothesis for physical crosslinking of the gel phase that is hydrophobically driven by pNIPAM. Theoretically, the LCST of the system is determined by the physicochemical properties of the pNIPAM polymer (e.g., molecular weight, end group), but it is also influenced by the environment (e.g., salt content). In the current study, PBS was introduced into the system; the transition temperature, therefore, was shifted to ~27 °C despite the fact that the coil-to-globule transition of pNIPAM-N_3_ (15 kDa) was reported to occur at ~30–32 °C. Compared to HA-L-pNIPAM_0.25_ and HA-L-pNIPAM_0.50_, HA-L-pNIPAM_0.10_ showed less significant changes in both moduli, which could be explained by the comparatively lower pNIPAM grafting density.

Based on these results, three candidates formulated from the synthetic HA-L-pNIPAM copolymers were proposed for further investigations into their dermatological application. Candidate 1 (2%, *w*/*w*) and Candidate 2 (2%, *w*/*w*) were prepared by directly dissolving HA-L-pNIPAM_0.10_ and HA-L-pNIPAM_0.25_ in PBS, Candidate 3 (2%, *w*/*w*) was a combination of HA-L-pNIPAM_0.50_ and non-derivatized linear HA (1500–1750 kDa) with a ratio of 3:1. Formulation preparation for the whole study strictly followed the method described previously since it was crucial to completely hydrate the polymer and to ensure the homogeneity of the hydrogel formulation. The HA content in each candidate formulation was calculated according to the DS values obtained from ^1^H NMR spectroscopy; it is displayed in Table 1.

The rheological properties of the three HA-L-pNIPAM hydrogel candidates were then determined at 22 °C and 37 °C with a constant oscillatory frequency of 0.7 Hz and shear stress set at 1.0 N/m^2^; Belotero Balance^®^ and Belotero Volume^®^ were used as references and were assessed under the same condition, but only at 37 °C. The values of G′, G″ and tan δ measured for the three candidates and two commercial products are presented in Table 2. Above the LCST (at 37 °C), Candidate 1 showed a G′ value in a range comparable to that of the reference Belotero Volume^®^, while Candidate 2 obtained an approximately twofold higher G′ value. Given that both candidates had a moderate G″ value, the tan δ (= $\frac{G^{\prime \prime }}{G^{\prime }}$) value of Candidates 1 and 2 were twofold (0.13 vs. 0.23) and threefold (0.08 vs. 0.23) lower than those of Belotero Volume^®^, respectively. In accordance with the measurements on the synthetic copolymers, the G′ values of Candidates 1 and 2 assessed at 22 °C (below the LCST) were significantly lower than those obtained at 37 °C (for the candidates and the commercial products) and resulted in increased tan δ values of 0.48 and 0.50, respectively.

These results suggested that Candidates 1 and 2 could be considered as realistic candidates for volumizing applications, and the comparatively high HA MW (1500–1750 kDa) employed in the synthesis of HA-L-pNIPAM copolymers (cf. 800 kDa in Belotero Balance^®^ [40]) contribute to their mechanical properties. On the one hand, the high G′ at 37 °C endowed the hydrogel with an excellent ability to regain its initial shape when the dynamic shearing forces were removed as reported by Gavard Molliard et al. [41], which corresponds to mechanical deformations such as muscle movements driving dynamic facial motion for a dermal filler in vivo. On the other, HA-L-pNIPAM hydrogel candidates with a comparatively low viscosity (G′ and tan δ value) at room temperature (22 °C) would require a lower extrusion force for injection, which would contribute to accurate and precise injections in the clinic. The spontaneous phase transition was also associated with Candidate 3, although it displayed viscoelastic characteristics that are more suitable for more superficial dermal injections, similar to Belotero Balance^®^, due to the presence of non-derivatized linear HA in the system. These results not only demonstrated the temperature-dependent characteristics of the proposed HA-L-pNIPAM hydrogel candidates, but also indicated that the products incorporating such HA-based polymers could be tailored by adjusting the HA MW, the DS, and the final concentration of the polymer to obtain defined viscoelastic properties for different applications. All the candidate formulations were stored at 37 °C and at 4 °C for 1 month; no significant change in rheological properties was observed.

#### 3.2.2. Injectability

The ease of injection is a critical parameter for all types of HA fillers, especially for volumizers. The force in Newtons (N) required to expel the hydrogel as a function of the stroke distance of the piston in a standard syringe with various needles is reported in Figure 2. All samples could be extruded through a 30G needle, Candidates 1–3 required a lower plateau force (i.e., <25 N) compared to the two reference products (i.e., 34 N for Belotero Balance^®^ and 42 N for Belotero Volume^®^). When a 32G needle was used for the injection, the mean force increased to close to 50 N for the three candidates (i.e., 49 N, 52 N and 51 N for Candidates 1, 2 and 3 respectively). Belotero Balance^®^ required a plateau force of 78 N, which represented a 2.3-fold increase in comparison with the 30G needle and 1.5 × more force than Candidates 1–3 with a 32G needle. Belotero Volume^®^ required more than 100 N and thus was considered not injectable in this case.

Given that a narrower gauge needle could potentially improve the precision of the injection and decrease the incidence of pain in clinical practice, notably where indications are expected to be painful (e.g., increase in lip volume), a 34G needle (external diameter 0.16 mm) representing a 50% thinner needle in comparison with a needle of 30G (external diameter 0.31 mm) was also tested. It was found that only Candidates 1–3 were able to be extruded through this thinner needle. Different injection forces were measured among the three, proportional to their viscoelastic values, meaning that Candidate 2 with higher values at room temperature required more force than Candidate 1, and Candidate 1 required more force than Candidate 3 (i.e., 72 N, 65 N and 55 N, respectively, as plateau force). Surprisingly, Candidate 3 did not show a significant difference between the 32G and the 34G (i.e., 51 N and 55 N respectively). Candidates 1 and 2 had a smoother horizontal plateau in the corresponding force injection profile. This aspect would contribute to facilitating the injection of the filler product and could be potentially linked to a more homogenous gel structure. Thus, a HA-based filler that reached clinically relevant values of G′ of more than 400 Pa was injected for the first time through a 34G needle.

#### 3.2.3. In Vitro Degradation

It is considered that dermal filler products are mainly degraded by oxidative stress, hyaluronidases, and mechanical stress in vivo [6,9,14,27]. Therefore, we compared HA-L-pNIPAM hydrogels to Belotero Balance^®^ with respect to/in terms of their enzymatic digestion and oxidative degradation at 37 °C in vitro. After the addition of hyaluronidases, much more striking effects were observed (Figure 3A–C). Belotero Balance^®^ and Candidates 1 and 2 followed the same decreasing trends in both moduli. Candidate 3 again displayed a viscoelastic behaviour with high standard deviations in the G′ and G″ values, as well as an unexpected sixfold increase in the G′ and an increase in the G″ values in the first 40 s followed by a decrease to 12.97 Pa by the end, leading to a significantly decreased tan (δ) value. It could be noticed that Belotero Balance^®^ yielded a tan (δ) of 1.15 in response to hyaluronidase exposure, which was twofold higher than those of Candidates 1–3, indicating a more fluid-like behaviour for the commercial reference. The storage modulus of Candidate 3 was lower than those of the other two candidates in the control group, in which 100 µL PBS was added to the candidate formulation instead of 100 µL H_2_O_2_/hyaluronidases); however, it yielded a value that was tenfold higher than those of all the other samples after contact with the enzyme for 10 min.

These results might be explained by the fact that Candidate 3 is a combination of raw linear HA and HA-L-pNIPAM_0.50_. The thermoresponsive behaviour of the highly grafted PNIPAM resulted in a change in conformation (hydrophobic interactions become dominant) above the LCST, which hypothetically created condensation and an entanglement of the HA chains, resulting in less available recognition sites (e.g., carboxylic acid groups for the enzyme). The conjugation of HA carboxyl groups and the formation of nanometric structures at body temperature are key elements to improve product resistance to hyaluronidases [33,34]. The higher DS on the carboxylic acid of HA chains resulted in a filling material that was more resistant with a superior ability to decrease the degradation rate. Meanwhile, after exposure to hydrogen peroxide (Figure 3D–F), rheological results showed a plateau for both moduli of Candidate 1 and Belotero Balance^®^, a slight increase in the G′ values and a slight decrease in the G″ values for Candidate 2. Therefore, tan (δ) values of the three samples were stable over the 10 min measurement period. However, Candidate 3 exhibited a completely different behaviour with a twofold increase in the G′ values and a 30% decrease in the G″ values, resulting in a significant decrease in the tan (δ) (from 0.82 to 0.32).

Overall, the accelerated degradation assay performed using hyaluronidases and hydrogen peroxide showed that hydrogel candidates presented similar or higher storage modulus (G′) values compared to those of Belotero Balance^®^ at the end of the measurements under both conditions, suggesting that the crosslinker-free technology can perform similarly to a typical crosslinked HA filler in terms of degradation.

### 3.3. Biological Evaluation of HA-L-pNIPAM Hydrogels

#### 3.3.1. In Vitro Biocompatibility on L929 Cell Line

Dermal fillers are categorized as a medical device that must undergo rigorous testing to determine its biocompatibility regardless of its mechanical, physical and chemical properties [42]. The biocompatibility of the non-derivatized linear HA (1500–1700 KDa) and the HA-L-pNIPAM hydrogel candidates was assessed in a mitochondrial activity assay using L929 mouse fibroblasts. The HA-L-pNIPAM hydrogel candidates were tested in the form of aqueous extract or their degradation product; cytocompatibility was measured by conducting WST-1 assays.

In the case of extracts (Figure 4), all the samples were not only proved to be non-toxic, but were also able to support cell proliferation because they had viability values > 100%. No significant differences in cell viability were found between 24 h and 48 h of culture (*p* > 0.05). For comparison, a 0.1% SDS (positive control) induced high levels of mortality (viability < 9%), while the diluted extraction samples (twofold and fourfold dilution) induced no cytotoxicity (see Appendix A). The degradation products of HA-L-pNIPAM hydrogels yielded viability values ~ 85% (Figure 4); cell viability was significantly higher in the presence of the degradation products of non-derivatized HA (*p* < 0.05).

The hypothesis is that the non-derivatized linear HA was efficiently digested by hyaluronidase as the enzyme had access to its backbone. In contrast, the spontaneous formation of nanostructure in the HA-L-pNIPAM hydrogels above the LCST (observed under SEM in a previous study [34]) hindered access to the cleavage sites and therefore limited the extent of enzyme digestion. As a result, cells in contact with these “partially digested” samples suffered from poor oxygen/nutrient diffusion, leading to a comparatively lower mitochondrial activity. These results showed that the HA-L-pNIPAM hydrogel candidates in their test forms (aqueous extract and enzyme degradation product) are compatible with L929 mouse fibroblasts. The test condition could be associated with the in vivo scenario considering the fact that dermal fillers were reported as being subject to swelling and enzymatic degradation after injection. The WST-1 assay was chosen to monitor the mitochondrial activity of cells after exposure to candidate formulations in this study since it is a relatively simple and cost-effective method. The assessment of the biocompatibility could be further completed by evaluating their effect on cell growth in terms of cell count, morphology, apoptosis, and cytokine profile.

#### 3.3.2. Tissue Integration in the Ex Vivo Porcine Skin Model

Porcine skin has been considered as a reliable surrogate for human skin given the similarities in skin structure and vascularization network, and this supports its use to predict histological behaviour in human skin after injection. Porcine ears were collected from a local slaughterhouse after the animal was sacrificed as food supply, respecting the *3R principles*—Replace, Reduce, Refine. Experimental setup of the ex vivo porcine skin model is presented in Appendix A. H&E-stained sections revealed an intact overall structure of skin explants harvested from the blank control group at T_0_ and T_72_; representative images obtained with the Axioscan Z1 brightfield microscope are presented in Appendix A. The main anatomical structures—stratum corneum, viable epidermis, epidermal–dermal junction—and appendages including hair follicles and secretory glands are well preserved after 3 days of cultivation; however, minor morphological changes could be noticed, e.g., hyperplasia—a wavy structure with an increased thickness of the epidermal layer—which is in accordance with previous reports on human explant skin culture that a general increase in epidermal thickness is noted over the first week in culture [43,44]. No further deterioration such as epidermal spongiosis (vacuolar fluid between the keratinocytes in the epidermis) or epidermal detachment (stratum basale separated from dermis) was observed in the epidermis despite it being considered more fragile than dermis due to its non-vascular structures and limited access to the nutrient supply in culture. Collagen bundles in the reticular dermis layer were seen to be slightly looser when comparing the samples at T_72_ to those at T_0_. The histological analysis demonstrated that tissue integrity is maintained unimpaired for up to 3 days in a culture under the described conditions; therefore, the ex vivo porcine explant skin model could be used for a short-term investigation.

Alcian Blue (AB) with PAS counterstaining [45,46,47] was employed to reveal the distribution pattern of the hydrogels and the way they entangle themselves with the surrounding adipocytes. Results of the validation performed with blank control group (no injection) and positive control group (subcutaneous injection of natural HA or Belotero Balance^®^) are presented in Appendix A. Representative AB/PAS-staining images of the investigated candidates at 0 and 72 h after subcutaneous injection are presented in Figure 5.

At T_0_, all the injected area appeared as an irregularly shaped depot with the material anchored in the fat lobules. The average dimensions of filler pools and filler spread patterns varied despite the fact that the same volume (50 µL) was injected in each case. This could result from the variation in the density of skin tissue at the injection site; therefore, the force required for injection would be influenced and the injected gel could receive forces/pressures from the surrounding tissues in a different pattern, which would be a factor leading to the variations in the swelling of the biomaterials. It could be noticed that Candidate 1 (Figure 5A) appears to be a continuous gel with a comparatively small size, while Candidate 2 (Figure 5B) shows larger granules and a continuous texture—both were distributed evenly throughout the subcutis as large homogeneous pools of HA material. However, a fibrous gel network embedded with particulates could be observed with Candidate 3 (Figure 5C), which exhibited the tendency to generate smaller pools and generously spread within the hypodermis. After incubation with the skin model for 72 h, all the injected materials were retained beneath the dermis at the injection site, but differences in the extent of degradation were observed—Candidate 1 (Figure 5D) maintained the depot of the HA materials with an excellent homogenous distribution, while a significant amount of Candidate 2 (Figure 5E) was degraded. The superior resistance of Candidate 3 (Figure 5F) could be attributed to the presence of linear HA. Our hypothesis is that the unmodified HA was fragmented while embedding in the tissue, but the network constructed by the HA-L-PNIPAM copolymer was conserved despite its uneven distribution pattern. These results could be related to the viscoelastic properties and in vitro degradation profile of the candidates, suggesting that Candidate 1 could be considered as the most promising dermal filling material.

Upon subcutaneous injection, Candidate 1 pushed against the adjacent adipocytes, and the displacement and compression of the soft fat lobules provided space for the HA filler integration into the subcutaneous fat, leading to the formation of a depot in the subcutis as shown in Figure 5A. However, a XY planar section of the skin explants (Figure 6A) showed that the fibrous trabecular network in the hypodermis was well preserved, and the injection bulks were constrained by the intercommunicating collagenous trabecular structures with limited movement. The material had a homogenous texture and was uniformly distributed, conforming to the adipose layer of the host tissue border without inducing any alteration in the cell morphology at both T_0_ (Figure 6B,C) and T_72_ (Figure 6D,E). Although a slight degradation of the materials was noticed at the interface with the host tissue after culture for 72 h, the homogenous filling material kept its uniform distribution pattern and was retained at the injection site. Interestingly, adipocytes appeared to be diffusing into the filling material in certain areas (Figure 6D), suggesting a good biocompatibility of the candidates and the skin tissue.

The interaction between the biological tissue and the injected filler was considered a critical criterion in the development of the HA dermal fillers. However, due to the new EU MDR requirements of prohibiting animal testing, an ex vivo porcine skin model developed in-house was employed for screening the innovative HA-L-pNIPAM hydrogels. In this study, the three HA-L-PNIPAM hydrogel candidates were injected into the hypodermis through a 30G needle that penetrated to a depth of approximately 4 mm as described previously. On the one hand, intradermal injection was reported as having a high risk of uneven distribution and host tissue response, as well as the formation of visible lumps under the skin [48]; on the other hand, a systematic imaging study on the subcutaneous distribution of the three commercially available HA filler formulations provided the validation of the excellent safety profile for subcutaneous injection, which was the approach in most actual clinical practice despite the product claims and indications [1,49]. AB/PAS-staining allowed for a direct visualization of the injected materials without any further step for labelling that might potentially change their behaviour. It not only provided evidence for the similarity of Candidate 1 to commercial dermal fillers in terms of biointegration immediately after injection (at T_0_) as previously reported [49], but also offered an insight into the dynamic behaviour of the candidates as a function of time in the subcutaneous skin layer under physiological conditions. It is of great interest to perform studies to quantitate biological responses in the ex vivo skin model to further understand the safety and efficacy of the proposed formulation.

Based on the original in vitro and ex vivo data presented herein, several future areas of focus have been identified, such as the development of a dermal filler product for lip enhancement or a body filler. Physicians use different types of HA to address various anatomic sites. The high physicochemical tunability and versatility of HA-L-pNIPAM technology may make it easy to develop specifications according to anatomic sites. A labial volumizer should exhibit a high shear modulus (i.e., exceeding 100 Pa) and strong cohesivity. Furthermore, it should be administered using extremely small needles which cause minimal discomfort, thereby facilitating clinical usage [41]. HA-L-pNIPAM candidates were tailored to reach a high storage modulus G′ at body temperature (i.e., up to 420 Pa) while remaining injectable through extremely fine 34G needles. Tissue integration into the hypodermis on an adapted ex vivo porcine skin model and a newly developed human abdominal skin model showed the hydrogel distribution pattern as a depot, with retention at the injection site. This innovation could enhance injection precision, reduce patient discomfort, and minimize post-injection hematoma. The potential formulations were not only established to be non-toxic and biocompatible, but also exhibited an extended residence duration at the site of injection due to the microgel structures of the system above the LCST [33,34].

This feature suggests that these formulations could be utilized as efficient drug delivery systems. They could enable extended and sustained release of therapeutic agents, thereby presenting a novel strategy for managing various dermatological conditions (e.g., psoriasis, herpes, acne, and alopecia). With this capability, they could potentially offer an innovative approach to treatment by providing the drugs where they are most needed and maintaining the therapeutic effect over a longer period. This enhanced delivery system could lead to better patient compliance and potentially improved outcomes in managing dermatological disorders.

## 4. Conclusions

In the present study, the advantages of HA-based thermosensitive fillers regarding hydrogel injectability, enzymatic degradation resistance and tissue integration were highlighted and compared to those of commercial reference products. Copolymer syntheses were performed in water; it should be possible to scale up the manufacturing process due to the water-based, robust scheme of the EDC/NHS amidation and the efficiency of the click chemistry approach. Three different fillers based on HA-L-pNIPAM (i.e., Candidates 1, 2 and 3) were tailored to reach a high storage modulus G′ at body temperature (i.e., from 92 Pa to 420 Pa) while remaining injectable through 34G needles at room temperature.

The ability of the thermosensitive fillers to form microgel structures above the LCST offers major benefits in comparison to chemically crosslinked HA fillers in terms of injectability for an easier clinical use including the increase in precision and pain reduction. The resistance of hydrogels to enzymatic and ROS-mediated degradation was higher or comparable to those of a commercial reference. A mix of linear HA 1.5–1.75 MDa and highly derived HA-L-PNIPAM_0.50_, Candidate 3, with HA contents of 9.5 mg/mL, showed a higher resistance to enzymatic degradation than the reference product, which was also observed after subcutaneous injection into an ex vivo porcine skin model developed in-house. Candidate 1 (HA-L-pNIPAM_0.10_, HA content of 15 mg/mL) was proposed as the most promising dermal filling material considering the balance of the HA content, viscoelastic properties, degradation profile and tissue integration.

The ex vivo and in vitro results presented herein warrant a next step toward in vivo studies to confirm the advantages of use including the volumizer properties of HA-L-pNIPAM formulations as a dermal filler product. The superior biocompatibility and excellent distribution pattern of these HA-L-pNIPAM hydrogels not only confirmed their potential as dermal fillers, but also opened the possibility for further investigation into their application as a drug delivery platform for a prolonged and sustained release of therapeutics. This is planned for future studies.

## Figures and Tables

**Figure 1 pharmaceutics-15-01708-f001:**
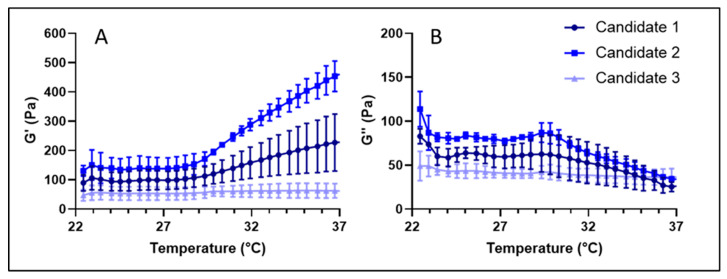
(**A**) G′ storage modulus and (**B**) G″ loss modulus of HA-L-pNIPAM hydrogel candidates as a function of temperature (n = 3, Mean ± SD).

**Figure 2 pharmaceutics-15-01708-f002:**
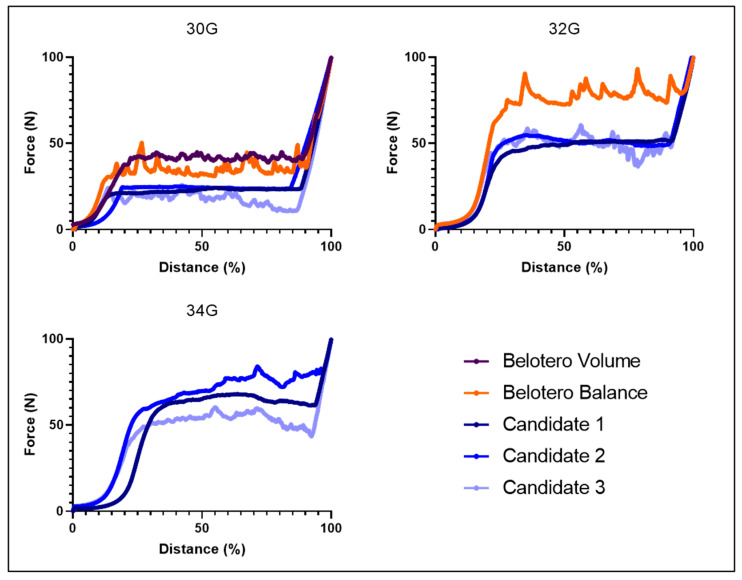
Force in Newtons (N) required to extrude Belotero Volume^®^, Belotero Balance^®^ and the three candidates as a function of the stroke distance of the piston in a syringe with a 30G, 32G and 34G needle (distance %) at 22 °C.

**Figure 3 pharmaceutics-15-01708-f003:**
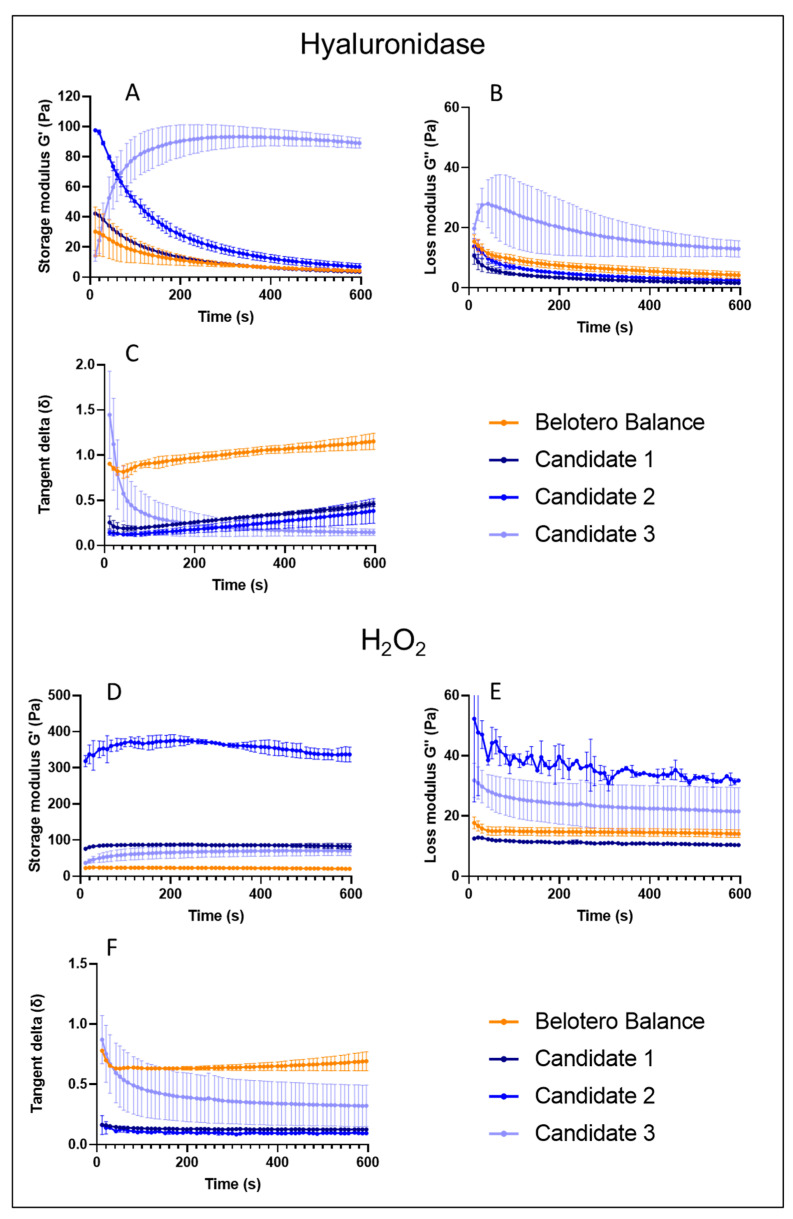
Belotero Balance^®^ and the three candidates storage modulus (G′), loss modulus (G″), and tangent delta (δ) normalized to initial values as function of time at 0.7 Hz and 37 °C after addition of 100 µL of hyaluronidase (100 U/mL) (**A**–**C**) (n = 3; Mean ± SD). After an addition of 100 µL H_2_O_2_ (30% *w*/*w*) (**D**–**F**) (n = 3; ±sd). As a control condition, 100 µL of PBS buffer was added to 400 µL of the sample.

**Figure 4 pharmaceutics-15-01708-f004:**
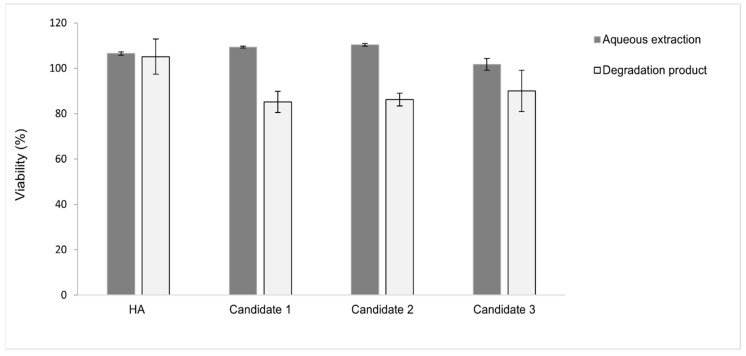
Effects of raw HA and HA-L-pNIPAM polymers on L929 cells at 24 h of exposure. Data express the percentage of cell viability with respect to culture media control (n = 3, Mean ± SD).

**Figure 5 pharmaceutics-15-01708-f005:**
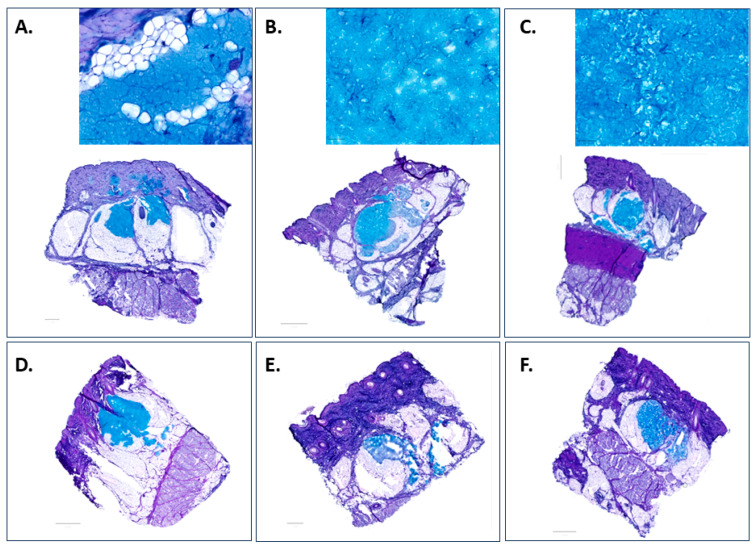
Comparison of subcutaneous injection of Candidates 1–3 in an ex vivo porcine skin model at T_0_ (**A**–**C**) and T_72_ (**D**–**F**). Light blue indicated the presence of the injected HA gel. Scale bar = 1 mm for (**A**,**E**) and 2 mm for (**B**–**D**,**F**).

**Figure 6 pharmaceutics-15-01708-f006:**
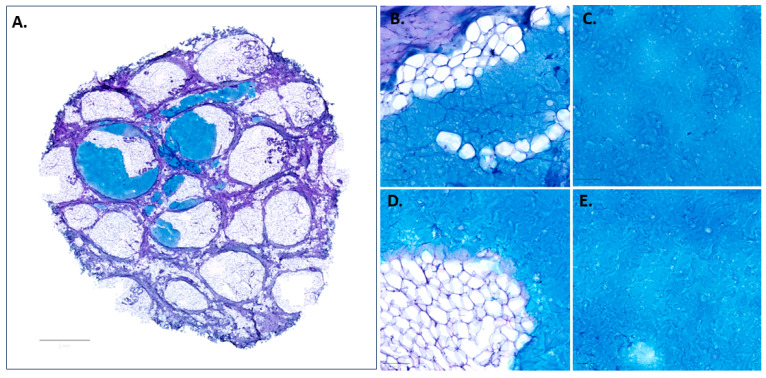
Distribution pattern and tissue integration of Candidate 1 at T_0_ (**A**–**C**) and T_72_ (**D**,**E**) in an ex vivo porcine skin model. Scale bar = 2 mm for (**A**), and 100 µm for (**B**–**E**).

**Table 1 pharmaceutics-15-01708-t001:** Summary of three HA-L-pNIPAM hydrogel candidates.

Formulation	Copolymer	Formulation Concentration (mg/mL)	HA Content (mg/mL)
Candidate 1	HA-L-pNIPAM_0.10_	20	15
Candidate 2	HA-L-pNIPAM_0.25_	20	8
Candidate 3	HA-L-pNIPAM_0.50_ + HA (1.5–1.75 MDa)	20	9.5

**Table 2 pharmaceutics-15-01708-t002:** Rheological properties of HA-L-PNIPAM hydrogel candidates and of commercial products (n = 3).

Formulation	HA Content (mg/mL)	Temperature (°C)	G′	G″	Tan δ = $\frac{G^{\prime \prime }}{G^{\prime }}$
Belotero Balance^®^	22.5	37	53	33	0.62
Belotero Volume^®^	26.0	37	233	54	0.23
Candidate 1	15.0	22	105.1	50.1	0.48
		37	234.1	28.7	0.13
Candidate 2	8.0	22	167.8	83.8	0.50
		37	420.2	34.9	0.08
Candidate 3	9.5	22	66.9	35.3	0.50
		37	92.5	43.5	0.49

## Data Availability

The data presented in this study are available on request from the corresponding author.

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
