# Peer review of "Injectable Hyaluronan-Based Thermoresponsive Hydrogels for Dermatological Applications"

_pharmaceutics, 2023, doi:10.3390/pharmaceutics15061708_

Round 1
Reviewer 1 Report
Gou et al. developed a thermosensitive dermal filler made by combining HA and a thermosensitive polymer, pNIPAAM. They propose a promising drug delivery system for treating skin and other disorders. Prior to publication, authors must address certain points to improve the quality of the work.
Abstract:
For example, the abstract is well written; however, authors may consider including more numerical data when presenting the experimental findings. For example, comparatively low viscosity at RT, superior resistance, lower injection force (compared to what), biocompatible (cell viability value?) etc. Additionally, the “green chemistry synthetic route” should be particularly emphasised in the abstract as the authors claimed that only water was used as a solvent in the polymer synthesis.
Keywords:
Keywords may be reconsidered. In my personal opinion, pNIPAM should be in the keywords. I am not entirely sure if the terms “physical properties and nanostructures” are suitable in the keywords. Instead of these terms, injectability and sustained drug delivery may be more suitable.
Introduction:
The introduction is nicely written. The authors should emphasise the significance of pNIPAM and how thermosensitive properties are essential in such a hydrogel design. It was stated that nanostructures are formed above a certain LCST value. The reasons that drive this formation should be explained. Also, what are the benefits of forming submicron spherical particles for such an application? Please mention them here. Additionally, the novelty of this work as well as the value added to the scientific literature should be addressed in the last paragraph.
The injectability of the hydrogel should be expanded in this section as the authors particularly investigated this property. For example, injectability at a different part of the body (i.e. under the eye) can be mentioned.
Material and Methods:
The methodology is clear and the authors explained the procedures in details.
Results and discussion:
In the results and discussion part, it seems like part 3.1., especially the part “the green chemistry synthetic routes for ………. to remove the residual pNIPAM” is supposed to be in the material and methods section. If the authors want to keep that information in the 3.1., they should enrich the section with more discussion as the current version seems to be a method.
Please specify the figure or table numbers in supporting information. For example, supporting information SI 2, not just supporting information.
In section 3.2.1. it says “Specific mechanical properties (e.g., stiffness, high G' values > 50 Pa) are required for injectable dermal 239 filling materials.” Please cite this statement.
The authors may consider measuring the exact LCST value of pNIPAM employing cloud point assay in addition to the theoretical calculation. It would also be interesting to measure the contact angle of the hydrogel (with/without pNIPAM) to confirm hydrophilicity and hydrophobicity below and above the LCST value. This may be important as the system was proposed by the authors to be used in sustained drug delivery.
Just asking out of curiosity, supposing you want to commercialise this product, the hydrogel should be stored in a certain condition. Does the storage time and condition affect the injectability? Do the authors have any comments on that? That would be an interesting discussion.
Having viability values over 100% does not necessarily mean it supports cell proliferation (Line 374). The degradation products yielded viability values, at some point close to 80%. Any discussion on how it may affect the in vivo applications in humans? Also, section 3.3.1 should be enriched with more discussion of the literature.
In section 3.3.2., I suggest the authors add the experimental setup for the porcine skin experiment and images of “porcine ear” before and after the injection (not only brightfield microscope images as shown in Figure 5). Besides, the reason to select a porcine ear should be given (why not a porcine belly, for instance?)
Please add scale bar information in the figure caption.
It may be something that I am missing but where are the SEM images of hydrogels above the LCST value?
Conclusion:
It was pointed out that the manufacturing process could be easily scaled up. Please discuss this more in this section. Is it because of the green synthetic route?
In the conclusion, when you mentioned Candidate 1 (line 484), expand its content in the parenthesis as it is a critical point. For example: (copolymer of HA-L-pNIPAM_0.10, HA content of 15 mg/mL etc).
Maybe a little bit more discussion on sustained drug release would be beneficial.
Overall, the quality of the English language is acceptable and the text is mostly clear.
Reviewer 2 Report
It was a manuscript about the synthesis and application of the hyaluronan-based thermoresponsive hydrogels used for dermatological applications. Here are some comments on this study that should be considered before publication:
1- The last paragraph of the introduction should be rewritten, it should contain a summary of information about the fabrication method and characterization.
2- “The green chemistry synthetic routes for the ...” Why do you mention green chemistry for the synthesis method?
3- Please mention the direction of each part related to the supporting information (such as Figure S1, ….).
4- “Compared to HA-L-pNIPAM0.25 and HA-L-pNIPAM0.50, HA-L-pNIPAM0.25 and HA-L-pNIPAM0.10 showed less significant …” Please rewrite this sentence, there is a mistake in it.
5- Please add the results of control condition to the results of degradation assay.
6- You didn’t mention the results of morphology and size in the results section.
7- Most of the references are out of date. Please update the references.
There are some grammatical and typo mistakes in the text that should be corrected, some of them are as follow:
· … minimally invasive procedures having been conducted in 2020 in the US alone …
· the storage modulus (G') and loss modulus (G") of … .
· …
Round 2
Reviewer 1 Report
The authors have addressed most of the comments. There are still minor points to be addressed prior to the publication.
The abstract and introduction have been improved. Only consider removing “i.e.,” after low viscosity in the abstract.
Please include the discussion regarding the storage conditions and stability in the manuscript: the product was stored for 1 month and 37 at +4 and no significant change in rheological properties was observed…” This statement was your response.
In the image showing the experimental setup with the porcine skin model, please use arrows to indicate which part is filler, and which part is porcine skin. For example, the white part is filler or fat? Because after a certain period of time, the fat part on the skin expands and covers most of the skin, which may be the case. So please use arrows to clearly indicate everything. It would be also great to include the moment you inject the filler with the needles.
Regarding the SEM images, if you don’t show the images in this study, no need to add morphological analysis in the material and method parts. So please remove that analysis from this study and just clearly cite the previous study and give a little bit of discussion on the particles.
Reviewer 2 Report
Thank you to the authors for their response, all the comments were responded, just as the last comment please delete section 2.3.4. from the method part.
